



# Applying machine learning for drought prediction using data from a large ensemble of climate simulations

Elizaveta Felsche[1,2,3] and Ralf Ludwig[2]

[1]Center for Digital Technology and Management, Munich, Germany
[2]Department of Geography, Ludwig Maximilians University of Munich, Munich, Germany
[3]Technical University of Munich, Munich, Germany

**Correspondence:** Elizaveta Felsche (felsche@cdtm.de)

**Abstract.**

There is strong scientific and social interest to understand the factors leading to extreme events in order to improve the management of risks associated with hazards like droughts. In this study, artificial neural networks are applied to predict the occurrence of a drought in two contrasting European domains, Munich and Lisbon, with a lead time of one month. The approach
takes into account a list of 30 atmospheric and soil variables as input parameters from a single-model initial condition large ensemble (CRCM5-LE). The data was produced the context of the ClimEx project by Ouranos with the Canadian Regional Climate Model (CRCM5) driven by 50 members of the Canadian Earth System Model (CanESM2). Drought occurrence was defined using the Standardized Precipitation Index. The best performing machine learning algorithms managed to obtain a correct classification of drought or no drought for a lead time of one month for around 55-60% of the events of each class for
both domains. Explainable AI methods like SHapley Additive exPlanations (SHAP) were applied to gain a better understanding of the trained algorithms. Variables like the North Atlantic Oscillation Index and air pressure one month before the event proved to be of high importance for the prediction. The study showed that seasonality has a high influence on goodness of drought prediction, especially for the Lisbon domain.

## 1   Introduction

Droughts remain to be one of the most dangerous hazards, having a serious and large-scale impact on environment, society and economy. Recent events like the Summer 2018 drought in huge parts of Central Europe led to severe forest fires and crop failures. The damage was estimated to several hundred millions euros solely in Germany (Federal Ministry of Food and Agriculture, 2018). Moreover the effect of global warming leads to major changes in the earth's climate system, having a direct influence on the frequency and severity of extreme events like droughts (Spinoni et al., 2016). An increase in frequency of
drought occurrence is a major threat for current and future generations, and comprehensive knowledge on the phenomenon of drought is needed in order to take action early and to prevent humanitarian catastrophes. This goes in conjunction with drought prediction. A tool for drought prediction would enable to mitigate the dangers connected to drought occurrences, such that it would advise stakeholders to store the maximal possible amount of water in the endangered regions. This would help to





mitigate the water shortage when the drought arrives. Measures for demand reduction could like that be introduced earlier, and

in better adjusted extent; this would help to reduce the economic and societal damage.

To mitigate the effects of droughts the information on the their onset is of crucial importance. This can be derived from a drought index. A variety of drought indices exist, which are typically defined according to statistical and physical measures. These are mostly taking into account atmospheric and soil variables. Among the most popular ones are the Standardized Precipitation Index (SPI), Standardized Precipitation Evaporation Index (SPEI), Soil Moisture Percentile (SMP), and Palmer

Drought Severity Index (PDSI). Standardized Precipitation Index (SPI) is adopted as the standard meteorological index by World Meteorological Organization (2012). It is a measure of meteorological drought based on the probability of occurrence of certain precipitation amounts in the area of interest (Sheffield and Wood, 2011). Studies on drought prediction by Belayneh et al. (2016) and Bonaccorso et al. (2015) use SPI as a prediction variable for the forecast.

Forecasting of any physical phenomenon can either be done by a physical, conceptual or data-driven model. The latter ones

are widely used due to their rapid development times and the flexibility in input parameters. McGovern et al. (2017) argues that AI-methods have a high potential for prediction of extremes due to the ability of machine learning methods to learn from past data, to handle large amounts of input variables, to integrate physical understanding into the models and to discover additional knowledge from the data.

A review on seasonal drought prediction given by Hao et al. (2018) identifies two typical predictor groups of variables:

large-scale climate indices that reflect the atmosphere-ocean circulation patterns and local climate variables. The first ones are known to correlate with the precipitation patterns in special regions and therefore are naturally correlated with the occurrence of drought. The teleconnection indices important for European precipitation include North Atlantic Oscillation (NAO), Scandinavian Oscillation (SCA), East Atlantic/Western Russia Oscillation (EA/WR), East Atlantic Oscillation (EA) and Atlantic Multidecadal Oscillation (AMO) (Hao et al., 2018). As shown by Folland et al. (2009) a positive NAO index in summer is asso-

ciated with dry and warm conditions in the north-west of Europe, whereas southern Europe and the Mediterranean experience cooler and wetter conditions. More information on the influence of the NAO, SCA, EA, and EAWR on the European climate can be found in Folland et al. (2009), Bueh and Nakamura (2007), Mikhailova and Yurovsky (2016), Lim (2015), Barnston and Livezey (1987) and Sheffield et al. (2009). A positive phase of AMO is associated with humid conditions over Great Britain and parts of Scandinavia and with dry conditions in the Mediterranean (Sheffield and Wood, 2011, p. 26); the negative phase

is associated with a reversed pattern: dry conditions in Great Britain and wet conditions in the Mediterranean. A study by Sheffield et al. (2009) showed a correlation between the amount of droughts and AMO of 62% with a significance at the 90% level. A recent study by Bonaccorso et al. (2015) uses NAO for prediction of probability of drought occurrence for Sicily. The local climate variables like precipitation, temperature, soil moisture were also used as inputs to reflect the conditions at the time the prediction occurs. Belayneh et al. (2016) and Bonaccorso et al. (2015) used SPI for the past months as input variable

to the algorithm. A study by Morid et al. (2007) used precipitation as an input parameter.

This paper examines the possibilities of meteorological drought prediction with the lead time of one month applying artificial neural networks (ANN) for two domains with different climate: one with Mediterranean (Lisbon), one with continental climate (Munich) (Ceglar et al., 2019). Both sites experienced an increase of drought frequency when comparing 2015 and 1950 and


are projected to keep rising under RCP4.5 as well as RCP 8.5. (Spinoni et al., 2017). Observational data offers only a limited

field for drought investigation as it can be seen from the following approximation. Systematical weather observations started in 1781 by Societas Meteorologica Palatina (Kington, 1980). In this study SPI1<-1 is used as a threshold for drought occurrence. It corresponds to the 15% driest months (John Keyantash, 2018) and can be estimated by a total amount of 430 events up to the year 2020 (Eq. 1).

$$(2020 - 1781) \text{ yr } \cdot 12 \text{ months/yr } \cdot 15\% = 430 \text{ events} \tag{1}$$

Compared to that CRCM5-LE offers a total amount of roughly 4500 events when using the first 50 years from the climate simulation data (1955-2005) (see Eq. 2).

$$50 \text{ yr/member } \cdot 50 \text{ members } \cdot 12 \text{ months/yr } \cdot 15\% = 4500 \text{ events} \tag{2}$$

This is a difference of an order of magnitude. The more data is available the better the predictions that can be derived by a drought predicting machine learning model and the more can be learned about drought formation. According to von Trentini

et al. (2020) precipitation in summer and winter derived from the European gridded data set (E-OBS) does fall to a high percentage into the range produced by CRCM5-LE for the historic period. Therefore, the CRCM5-LE proves applicable to this study and its larger amount of extreme events can be used as input to the machine learning algorithms. In this study a variety of ANNs were trained. Best performing models were investigated to using explainable AI methods to understand the results.

While no comparable study exists for the Munich domain, Santos et al. (2014) performed a drought prediction based on

SPI6 for Portugal for the months April, May and June using the following input variables: sea surface temperatures (JFM), NAO (DJFM) and cumulative precipitation (NDJFM for $SPI6_{April}$, DJFM for $SPI6_{May}$, JFM for $SPI6_{June}$). Best results were achieved for the prediction of SPI6 for April with a correlation coefficient of 0.98. SPI6 for May and June referred to a correlation coefficient of 0.78 and 0.77 respectively.

## 2 Data and Methods

### 2.1 Datasets

To investigate the predictability of droughts data from the single-model initial condition large ensemble (SMILE) consisting of 50 members, the Canadian Regional Climate Model 5 Large Ensemble (CRCM5-LE) was used. The data was produced within the scope of the ClimEx Project (Leduc et al. (2019), www.climex-project.org). The CRCM5-LE was generated by dynamical downscaling of the data provided by the 50-member initial condition Canadian Earth Sysytem Model 2 using the Canadian

Regional Climate Model 5 (Martynov et al., 2013). The data has a resolution of $0.11 \deg$ (12 km) and is produced for the years 1950-2099 for a European and an eastern North America domain. For the years 1950-2005 the historical green house gas concentrations and aerosol emissions are being used. Starting from 2005, the model introduces the RCP8.5 (IPCC, 2013) forcing scenario. A total of 41 atmospheric variables is available in a temporal resolution of one to three hours. They are used on monthly basis as input to the machine learning algorithm. The list of variables is provided in Tab. 1.



| clt | Total Cloud Fraction | % | ps | Surface Air Pressure | Pa |
|-----|---------------------|---|-----|---------------------|-----|
| dds | Near-Surface Dewpoint Depression | K | psl | Sea Level Pressure | Pa |
| evspsbl | Evaporation | $kgm^{-2}s^{-1}$ | rlds | Surface Downwelling Longwave Radiation | $Wm^{-2}$ |
| evspsblland | Water Evaporation from Land | $kgm^{-2}s^{-1}$ | rlus | Surface Upwelling Longwave Radiation | $Wm^{-2}$ |
| hfls | Surface Upward Latent Heat Flux | $Wm^{-2}$ | rlut | TOA Outgoing Longwave Radiation | $Wm^{-2}$ |
| hfss | Surface Upward Sensible Heat Flux | $Wm^{-2}$ | rsaa | Shortwave Radiation Absorbed by Atmosphere | $Wm^{-2}$ |
| hurs | Near-Surface Relative Humidity | % | rsds | Surface Downwelling Shortwave Radiation | $Wm^{-2}$ |
| huss | Near-Surface Specific Humidity | 1 | rsdt | TOA Incident Shortwave Radiation | $Wm^{-2}$ |
| mrfso | Soil Frozen Water Content | $kgm^{-2}$ | rsus | Surface Upwelling Shortwave Radiation | $Wm^{-2}$ |
| mrro | Total Runoff | $kgm^{-2}s-1$ | rsut | TOA Outgoing Shortwave Radiation | $Wm^{-2}$ |
| mrros | Surface Runoff | $kgm^{-2}s^{-1}$ | sfcWindmax | Daily Maximum Near-Surface Wind Speed | $ms^{-1}$ |
| mrso | Total Soil Moisture Content | $kgm^{-2}$ | snc | Snow Area Fraction | % |
| mrsos | Moisture in Upper Portion of Soil Column | $kgm^{-2}$ | snd | Snow Depth | m |
| prc | Convective Precipitation | $kgm^{-2}s^{-1}$ | snw | Surface Snow Amount | $kgm^{-2}$ |
| prdc | Deep Convective Precipitation | $kgm^{-2}s^{-1}$ | tas | Near-Surface Air Temperature | K |
| prfr | Freezing Rain | $kgm^{-2}s^{-1}$ | tasmax | Daily Maximum Near-Surface Temperature | K |
| pr | Precipitaiton | $kgm^{-2}s^{-1}$ | tasmin | Daily Minimum Near-Surface Temperature | K |
| prlp | Liquid Precipitation | $kgm^{-2}s^{-1}$ | ts | Surface Temperature | K |
| prrp | Refrozen Rain | $kgm^{-2}s^{-1}$ | uas | Eastward Near-Surface Wind | $ms^{-1}$ |
| prsn | Snowfall Flux | $kgm^{-2}s^{-1}$ | vas | Northward Near-Surface Wind | $ms^{-1}$ |
| prw | Water Vapor Path | $kgm^{-2}$ | | | |

**Table 1.** Monthly variables from CRCM5-LE

This study uses the monthly sea level pressure (*pr*) from the CanESM2-LE (Kushner et al., 2018; Kirchmeier-Young et al., 2016) for the calculation of North Atlantic Oscillation (NAO), Scandinavian Oscillation (SCA), East Atlantic Oscillation (EA) and East Atlantic/Western Russia Oscillation (EA/WR) over the whole Atlantic basin ($20°-80°$N, $90°$W$-40°$E). The Atlantic Multidecadal Oscillation (AMO) was calculated using the Sea Surface Temperature (SST) over the $0-60°$N, $0-80°$W from the CanESM2. Only the period 1955-2005 was considered in order to stay within the scope of historical climate. The CRCM5 domain is displayed in Fig. 1. For the machine learning training a gridpoint situated as $48.11°$N and $-9.17°$W is referenced as Munich and $38.67°$N and $11.91°$W is referenced as Lisbon.

## 2.2 Input variables for drought prediction

In order to calculate NAO, SCA, EA and EA/WR the method introduced by Hurrell et al. (2003) was used: a principal component analysis (PCA) of the monthly sea level pressure variables was performed over the $20°-80°$N, $90°$W$-40°$E domain. The leading eigenvectors, scaled by the amount of variance they explain, represent the leading circulation patterns of the atmospheric system. The first eigenvector corresponds to NAO, the second one to SCA, the third one to EA, the fourth one to EA/WR. To calculate the teleconnection indices (NAO, SCA, EA, EA/WR) the Eof package described in Dawson (2016) was used. The leading modes of the PCA corresponding to NAO, SCA, EA and EA/WR derived from the CanEsm2 dataset are shown in Fig. 2.

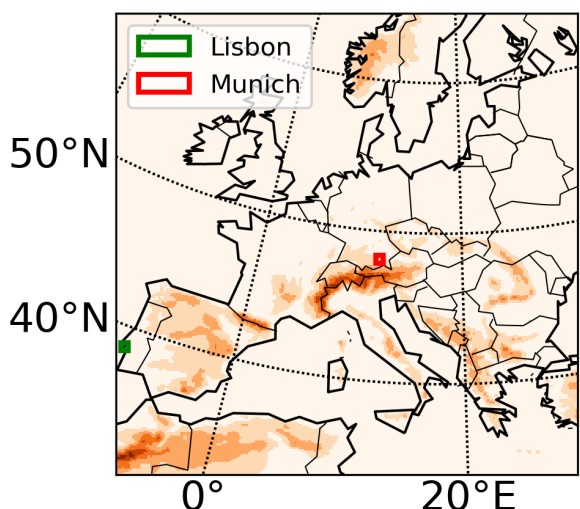

**Figure 1.** CRCM5 topography.

AMO is calculated by spatial averaging over the $0-60°\mathrm{N}, 0-80°\mathrm{W}$ area of the anomaly of sea surface temperature (E Trenberth, 2011). Additionally the 10-year running mean of AMO is calculated as an input variable, as it is widely used in various studies and was shown to be correlated with precipitation (Enfield et al., 2001).

To limit the computation time a preselection of variables for the input data was performed. In order to eliminate redundant variables Pearsons R between the CRCM5 variables for the chosen domains was calculated. Pearsons R ($\rho_{X,Y}$) is a measure of linear correlation between two variables X and Y. $\rho_{X,Y}$ equals 1 if the correlation is total positive, 0 if there is no linear correlation and -1 if the correlation is total negative (Guyon and Elisseeff, 2003). For two samples $x$ and $y$ the Pearsons R is defined in the following way:

$$\rho_{x,y} = \frac{\sum_{i=1}^n (x_i - \bar{x})(y_i - \bar{y})}{\sqrt{\sum_{i=1}^n (x_i - \bar{x})^2}\sqrt{\sum_{i=1}^n (y_i - \bar{y})^2}} \tag{3}$$

The bar refers to the average over the index $i$ (Guyon and Elisseeff, 2003). $\rho$ was calculated for all possible permutations of the 41 input variables. The ones correlating to a high degree were examined and a threshold of 0.95 was chosen. In Tab. 2 a list of sorted out variables and the corresponding values of Pearson R is given. The high correlation values can be explained by a physical relationship between the variables: e.g. the total evaporation (*evpsbl*) is almost the same as evaporation from land (*evspsblland*), as there are no relevant water bodies in the chosen domains. Out of the full list of 41 variables 14 were sorted out as being redundant.

## 2.3 Standardized Precipitation Index

The Standardized Precipitation Index (SPI) is a precipitation based index introduced by McKee et al. (1993). For the calculation of SPI a continuous monthly precipitation dataset is used. The index can be calculated on different timescales: typically, it is


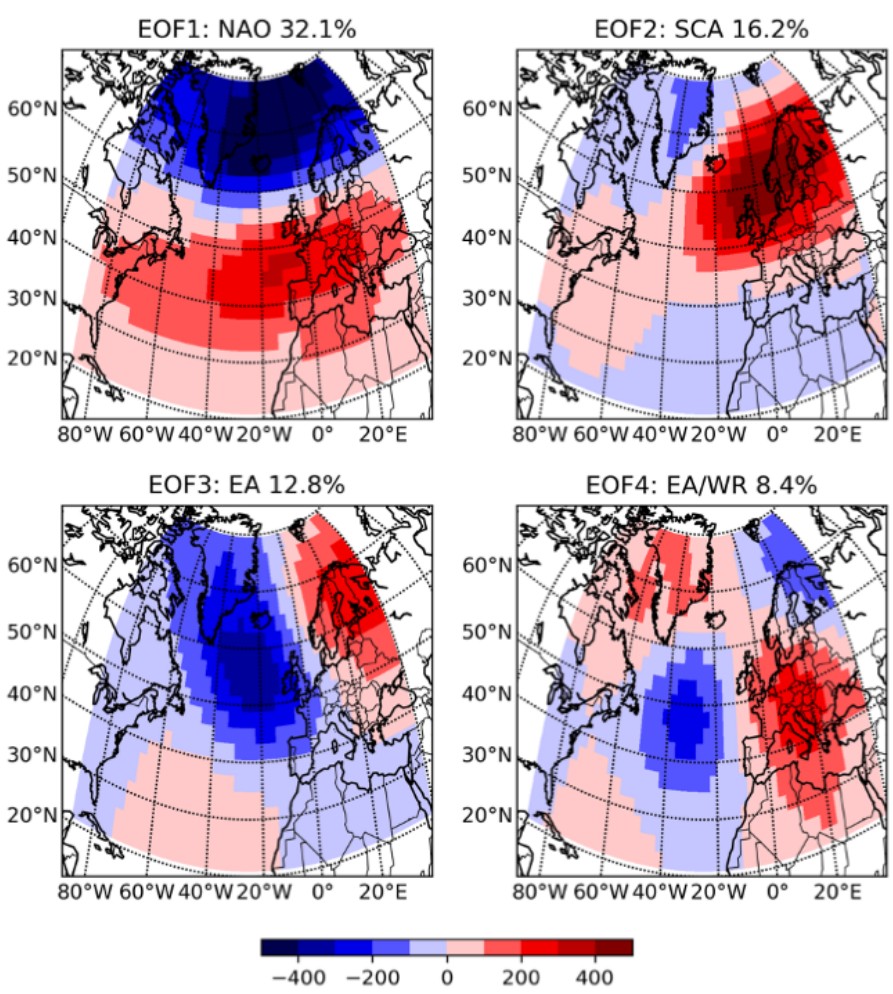

**Figure 2.** First four leading eigenfunctions of the mean sea level pressure in CanESM2. Percentage of variance the mode explains is given on top of the figures.





| Kept variable | Sorted out variable | Pearsons R |
|---|---|---|
| *hurs* | *dds* | -0.9879 |
| *evspsbl* | *evspsblland* | 0.9994 |
| *evspsbl* | *hfls* | 0.9988 |
| *mrso* | *mrlso* | 0.9991 |
| *rlut* | *rlaa* | -0.9549 |
| *tas* | *rlds* | 0.9550 |
| *tas* | *rlus* | 0.9960 |
| *rnt* | *rns* | 0.9954 |
| *rnt* | *rsaa* | 0.9831 |
| *rnt* | *rsdt* | 0.9970 |
| *rnt* | *rss* | 0.9872 |
| *rnt* | *rst* | 0.9926 |
| *tas* | *tasmax* | 0.9932 |
| *tas* | *tasmin* | 0.9864 |

**Table 2.** List of sorted out variables

1, 3, 6, 12 or 24 months. As a first step the precipitation values are accumulated for the needed timescale. The resulting dataset is fitted to a Gamma distribution for each month separately and then transformed to a normal distribution, such that the mean SPI is zero. The SPI value for a given precipitation is then the number of standard deviations from normal. Because of the normalization SPI is especially useful to represent wetter and drier climates, as well as to account for differences among seasons. Here SPI1 was calculated for Lisbon and Munich each using the data from 1955-2005 from all members as reference.

## 2.4 Machine learning

This study investigates drought predictability applying the technique of supervised machine learning for this purpose. Machine learning is a promising tool for the analysis of complex and data-rich phenomena as droughts (McGovern et al., 2017). The python package Keras, a high-level neural network package, was used for the design of the machine learning models (Chollet et al., 2015), as it allows to design neural networks in an easy way by adding layers. Three crucial elements are needed to perform drought prediction by supervised machine learning: input data, a target variable to be predicted and a computation pipeline, which includes the machine learning algorithm.

The data from the years 1957 - 1999 was used as training data, the years 2000-2005 were used for the testing purpose. A small fraction of the training data was used for the validation of the machine learning algorithms. The target variable chosen for the prediction of droughts was SPI. Two classes for the prediction were identified in the following way: $SPI1 < -1$ was defined as an event and was initialized with 1, $SPI1 > -1$ was initialized with 0 and corresponded to a non-drought event. The lead time of one month was chosen for the prediction





After the feature selection 27 variables originating directly from the CRCM5-LE dataset were used as input, each of them in

a timeseries of 12 months. In addition to those the teleconnection indices NAO, SCA, EA, EA/WR, AMO and AMO10 were used as input.

For this analysis we used a supervised machine learning algorithm, an Artificial Neural Network (ANN). ANNs are algorithms whose design is inspired by the architecture of the human brain with its neurons (Russell and Norvig, 2009).; they both consists of connected nodes. A link between the node $i$ and the node $j$ serves to propagate the activation $a_i$ from $i$ to $j$. To each

connection a numeric weight $w_{i,j}$ is assigned. The output of the node is computed by:

$$a_i = g(in_j) = g\left(\sum_{i=0}^{n} w_{i,j} a_i\right) \tag{4}$$

(Russell and Norvig, 2009, p. 728). The activation function defines the output of the node. In order to have stable learners with confident predictions a function with a soft threshold is recommended (Russell and Norvig, 2009). In this study the following three activation functions were used: Sigmoid, Rectified Linear Unit (ReLU), Exponential Linear Unit (ELU). Sigmoid acti-

vation is especially useful for the output layer (Russell and Norvig, 2009), while ReLU and ELU both have the property of allowing very fast optimization (Maas, 2013) .

*Sigmoid* function, also called logistic function, is defined in the following way:

$$Logistic(x) = \frac{1}{1 + e^{-x}} \tag{5}$$

(Russell and Norvig, 2009). This function has an output between 0 and 1. This can be interpreted as a probability of belonging

to the class 1. One of the main disadvantages of the sigmoid activation function is the vanishing gradient problem: at higher, almost saturated layers with values of 1 or -1, the gradients become nearly 0 resulting in a slow optimization convergence (Russell and Norvig, 2009, p. 726).

*ReLU* refers to Rectified Linear Unit and shows better performance when dealing with the vanishing gradient problem (Maas, 2013). ReLU is defined in the following way:

$$f(x) = max(0, x) \tag{6}$$

*ELU* refers to the Exponential Linear Unit and was introduced by Clevert et al. (2016). Clevert et al. (2016) claim that in experiments the ELU activation led to faster learning and significantly better generalization performance than ReLU and sigmoid activation. The function is defined as:

$$f(x) = \begin{cases} x \text{ if } x > 0 \\ \alpha(exp(x) - 1) \text{ if } x \leq 0 \end{cases} \tag{7}$$

$\alpha$ controls the value to which an ELU saturates for negative inputs. Per default the value is set to 1 such that the function saturates at -1.

Two kinds of layers were used in this study: Dense and Dropout. *Dense* refers to a regular fully connected neural network layer. *Dropout* refers to a layer which is randomly setting a fraction of inputs to zero at each update. This technique is used to





prevent overfitting and therefore improving the performance of the algorithm (Chollet et al., 2015). The first part of the study

concentrated on the methodological search for the best performing algorithms. A pipeline to search for the best performing architecture, value for L2 regularization and loss function was built up.

   The model performance was evaluated using Accuracy and F1-score (Sasaki, 2007). The latter one is especially useful when training on datasets with an imbalanced class distribution, as it is in the case of our dataset. Accuracy is defined in the following way:

$$Accuracy = \frac{Number\ of\ right\ predictions}{Total\ number\ of\ samples} \tag{8}$$

   F1-score is a harmonic measure between precision and recall. Precision is the amount of true positives with respect to the amount of positively classified data. Recall is the amount of true positives with respect to the total number of positives in the data. F1-score is defined in the following way:

$$F1 - score = 2\frac{Precision \cdot Recall}{Precision + Recall} \tag{9}$$

Due to the class imbalance within the dataset we require that the accuracy on each class is at least 50%. In that case given the distribution of the test dataset of 1803 non-drought events to 387 droughts for Lisbon and 1848 non-drought events to 352 drought events for Munich a marginal F1-score of 0.26 for Lisbon and 0.24 for Munich is given.

   The second part of the study analyzed the best performing algorithms (one for Lisbon domain, one for Munich domain) by applying explainable AI methods. SHAP (SHapley Additive exPlanations) is a state of the art method for interpretation of

machine learning models, which was inspired by game theory (Lundberg and Lee, 2017). It estimates for each input feature an average marginal contribution to the prediction of the result and therefore allows a comparison of the contributions among different features. In addition to that the difference in predictability among the seasons is calculated and compared to gain a better understanding on the influence of seasonal weather patterns.

## 3 Results

This study consists of two parts: the first parts deals with a systematical search for the best performing setup of the ANN model for the two domains of interest: Munich and Lisbon. A repeated training was conducted by varying the values of parameters like the architecture of the hidden layers, L2 Regularization and the loss function. In the second part of the analysis the best performing models for the two domains were analyzed using explainable AI methods.

### 3.1 Model training results

For the design of the ANN it is crucial to perform fine tuning of the model parameters to find the optimal setup. An architecture has to have enough layers and neurons to capture the complexity of the dataset (Goodfellow et al., 2016). In order to find the best architecture the learning curve of the algorithm was inspected. The learning curve shows the loss of the training and validation datasets on the weights during the training (Goodfellow et al., 2016). Two examples are shown in figure 3. The




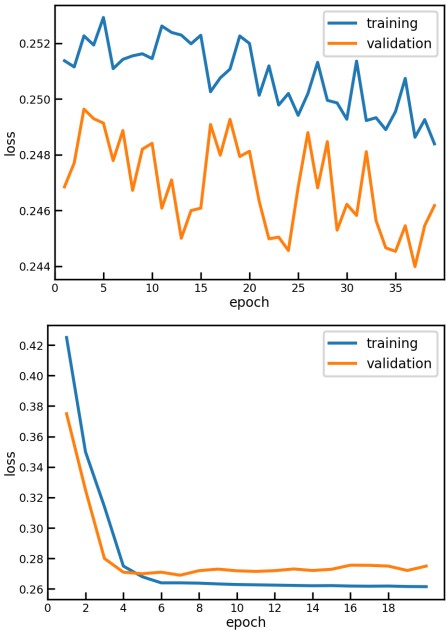

**Figure 3.** Learning curve for two chosen fitting examples: algorithm complexity insufficient (top) and overfitting (bottom)

plot shown in the top refers to an architecture, which is not able to capture the complexity of the dataset: the loss is hardly decreasing on the training or validation data. The bottom figure refers to an architecture which overfits: in the last epochs the loss of the validation dataset is rising, while it decreases on the training dataset.

In such way a network was searched which captured the given complexity of the dataset. This was reached with an algorithm consisting of at least five layers. Additionally two dropout layers, which are setting a specified number of nodes to zero in a random way, were introduced in order to fight overfitting.

### 3.1.1 L2-Regularization

L2-Regularization is a broadly applied method to prevent overfitting on the training data (Bishop, 2007). The main idea behind regularization is to add a penalty term to the loss function, which will punish the classifier for complexity and force some of the weights to zero (Russell and Norvig, 2009). In case of L2 regularization the punishing term is proportional to L2-norm of the weight vector. The weight of the punishing term $\lambda$ determines the relative importance of the regularization.

The results of the training with different values of $\lambda$ for L2 regularization are shown in Tab. 3. Training results are displayed in this particular case since the regularization is introduced to prevent overfitting. Generally the performance on the test dataset is more important and will be inspected in following experiments. If $\lambda$ is set to zero the regularization term vanishes. Especially in those cases the overfitting is high. For Lisbon overall higher performance could be seen for values of $\lambda$ around 0.01, 0.001 or 0.0001. Models that were trained on the Munich dataset performed better with the value of $\lambda$ of 0.001. Since the performance





of the model on the F1-score has a higher importance for an imbalanced dataset than the pure accuracy the value of 0.001 was chosen for the following ANN model training.

| | Lisbon | | | | Munich | | | |
| | Train | | Test | | Train | | Test | |
| λ | Acc | F1 | Acc | F1 | Acc | F1 | Acc | F1 |
|---|---|---|---|---|---|---|---|---|
| 0 | 0.961 | 0.861 | 0.733 | 0.206 | 0.959 | 0.865 | 0.787 | 0.176 |
| 0.1 | 0.495 | 0.233 | 0.373 | 0.294 | 0.506 | 0.241 | 0.536 | 0.215 |
| 0.01 | 0.517 | 0.245 | 0.460 | 0.269 | 0.519 | 0.268 | 0.431 | 0.275 |
| 0.001 | 0.572 | 0.261 | 0.540 | 0.288 | 0.490 | 0.288 | 0.563 | 0.266 |
| 0.0001 | 0.765 | 0.472 | 0.627 | 0.259 | 0.823 | 0.557 | 0.719 | 0.189 |

**Table 3.** Results of ANN training for different values for λ for L2 regularization. Best performing algorithm, defined as the one with an accuracy of at least 50 % and the highest F1-Score, shown in light grey.

### 3.1.2 Loss function

As a next step the influence of the different loss functions on the model performance was investigated. Loss function is a function to evaluate how well a specific algorithm manages to fit the training data (Janocha and Czarnecki, 2017). It is an
important part of the optimization function which has a direct influence on the updating of the weights of the ANN (Russell and Norvig, 2009). In addition to overall accuracy and F1-metric, the accuracies on the non-drought and drought classes are displayed. The results are shown in Tab. 4. Binary cross-entropy, mean absolute error and hinge loss functions showed the best performance for the Munich domain. In contrast to that for the Lisbon domain only the mean absolute error loss function had an accuracy of higher than 0.5. Also in the case of the Munich domain mean absolute error showed a higher performance on
the F1-score. Therefore mean absolute error it is used for further analysis.

| | Lisbon | | | | Munich | | | |
| Loss-Function | Acc nd | Acc d | Acc | F1 | Acc nd | Acc d | Acc | F1 |
|---|---|---|---|---|---|---|---|---|
| mean absolute error | 0.511 | 0.516 | 0.540 | 0.288 | 0.500 | 0.582 | 0.512 | 0.276 |
| mean squared error | 0.440 | 0.655 | 0.479 | 0.312 | 0.562 | 0.509 | 0.553 | 0.267 |
| binary crossentropy | 0.436 | 0.610 | 0.467 | 0.292 | 0.589 | 0.440 | 0.565 | 0.245 |
| hinge | 0.229 | 0.753 | 0.323 | 0.287 | 0.568 | 0.486 | 0.555 | 0.259 |
| squared hinge | 0.486 | 0.501 | 0.489 | 0.261 | 1.000 | 0.000 | 0.840 | 0.000 |

**Table 4.** Training results for different loss functions. Acc nd refers to the accuracy on the non-drought class and Acc d to the accuracy of the drought class




### 3.1.3 Model architecture

Lastly the models were trained on both domains using different architectures. Table 5 is displaying the training results. The column "architecture" refers to the number of neurons in each Dense (De) layer separated by the *-sign. For Dropout (Dr) layers the fraction of weights which were randomly set to zero is given. We require the accuracy on both classes to be higher

than 0.5 and search for an F1-score as high as possible. In case of the Lisbon domain three trained models are satisfying the criterion of at least 50% accuracy on each class: the model in the first, in the fourth and in the last row. Best performance in terms of F1-score is obtained for the last model with the following architecture: 5000*0.5*4000*0.5*1000*500*100. For the Munich domain only the first and the fourth models are satisfying the criterion of at least 50% accuracy on each class. For further analyses the first model is chosen, as it shows the highest F1-score. The following model architecture was used for the

Munich domain: 4000*0.5*1000*0.5*500*100*5. In the next step those models were analyzed using explainable AI methods.

| | | Lisbon | | | | Munich | | | |
|---|---|---|---|---|---|---|---|---|---|
| Neurons | Architecture | Acc nd | Acc d | Acc | F1 | Acc nd | Acc d | Acc | F1 |
| De*Dr*De*Dr*De*De*De | 4000*0.5*1000*0.5*500*100*5 | 0.511 | 0.516 | 0.540 | 0.288 | 0.562 | 0.509 | 0.553 | 0.267 |
| De*Dr*De*Dr*De*De*De | 5000*0.5*1000*0.5*500*100*5 | 0.581 | 0.496 | 0.566 | 0.292 | 0.378 | 0.693 | 0.428 | 0.279 |
| De*Dr*De*Dr*De*De*De | 5000*0.5*4000*0.5*500*100*5 | 0.457 | 0.602 | 0.483 | 0.296 | 0.725 | 0.338 | 0.663 | 0.243 |
| De*Dr*De*Dr*De*De*De | 5000*0.5*4000*0.5*1000*100*5 | 0.570 | 0.501 | 0.558 | 0.290 | 0.527 | 0.514 | 0.525 | 0.257 |
| De*Dr*De*Dr*De*De*De | 5000*0.5*4000*0.5*1000*500*5 | 0.402 | 0.635 | 0.444 | 0.292 | 0.683 | 0.409 | 0.640 | 0.266 |
| De*Dr*De*Dr*De*De*De | 5000*0.5*4000*0.5*1000*500*100 | 0.575 | 0.526 | 0.566 | 0.305 | 0.420 | 0.619 | 0.452 | 0.266 |

**Table 5.** Training results for the Lisbon and Munich domains for different variations of architecture. Best results marked in gray. Acc nd refers to the accuracy on the non-drought class and Acc d to the accuracy of the drought class

## 3.2 Explainable AI methods for the analysis of best performing algorithms

### 3.2.1 Shapely values

For the Munich and Lisbon domain Shapely values were calculated for the best performing models. The results are shown in

Fig. 4. Since the calculation of Shapely values is computationally expensive they were calculated 5 times on a subset of 500 data points. The error bars displayed in black on the plot indicate that the uncertainties are smaller than the nominal values of the variable contributions. The nominal Shapely values were normed and recalculated to a percentage of contribution to the prediction, e.g. the NAO1 value explains roughly 2.3% of the prediction for the Lisbon domain. We see that for both domains the contribution to the prediction is broadly distributed among the many input variables. Between Lisbon and Munich Shapely

values show a distinct difference in the nominal values of the feature contributions: values for Lisbon are about 6 times higher than those for Munich (e. g. the contribution of NAO1 for Munich is around 0.3% and for Lisbon around 1.9%).



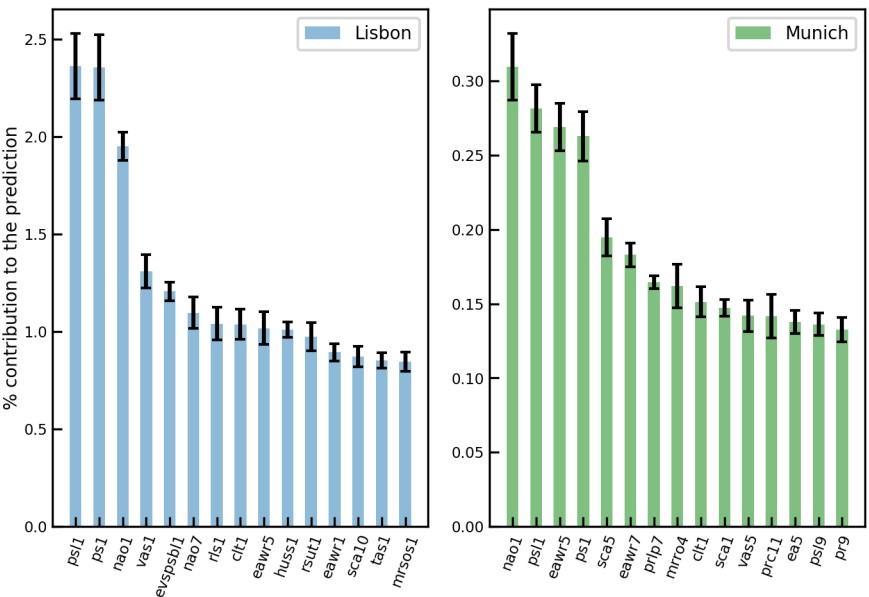

**Figure 4.** Mean Shapely values normalized to the contribution to the prediction for the top 15 variables with the highest importance for Lisbon (left) and Munich (right). The number behind the variable name refers to the number of months before the event (NAO1 - NAO value one month before the predicted event).

For the Lisbon domain, the variables with a higher impact are sea level pressure (*psl*), surface pressure (*ps*) and NAO one month before the event. The first two variables are strongly autocorrelated for the Lisbon domain due to its location at the sea. The stong influence of *ps/psl* and NAO shows the influence of the atmospheric pressure system on drought formation in Lisbon. It is also striking that the influence of the local pressure seems to be higher than the influence of NAO. The next two variables for the Lisbon domain with the strongest contribution to the prediction are Northward Near-Surface Wind (*vas*) and Evaporation (*evspsbl*). The latter variable has a very direct influence on the formation of drought given that if evaporation is getting lower, also the probability of formation of rain clouds decreases (Sheffield and Wood, 2011). The contribution of *vas* to drought formation in Lisbon needs to be further studied. For the Munich domain the highest influence is found for NAO1, *psl1* , EAWR5 and *ps*. The results indicate that NAO is the most influential drought predictor for Munich. Additionally the contribution of EAWR5 and SCA5 on the Munich domain cannot be neglected as they are found within the top five predictors. A further investigation of this relationship is of interest for the understanding of drought formation in Munich.

### 3.2.2 Seasonality

In order to evaluate the influence of seasonality on the prediction the performance of the model was calculated separately for the four seasons. Since the distribution between the drought and non-drought classes was different among the seasons (e.g. range of 17% to 19% of drought events for the Lisbon domain) a rescaling of the number of drought and non-drought events


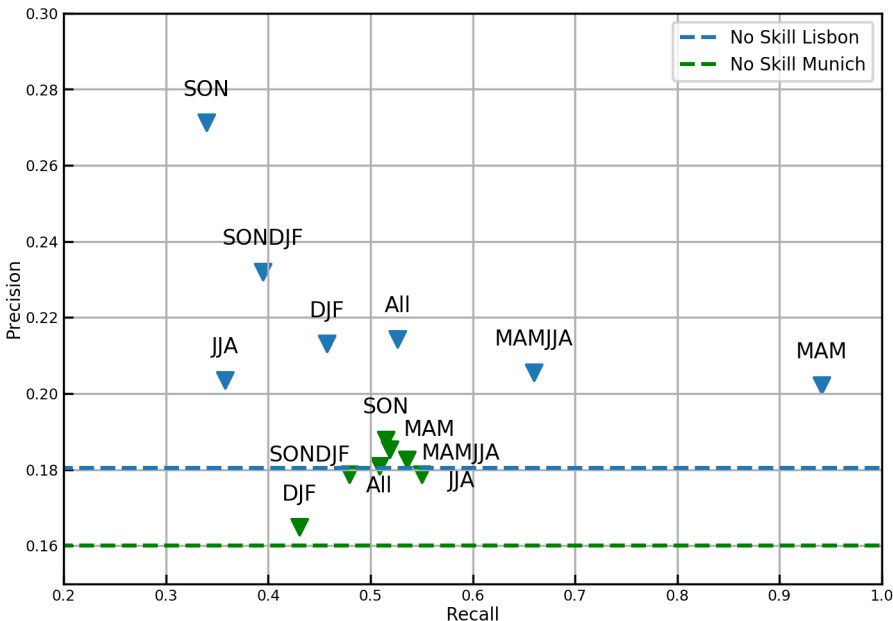

**Figure 5.** The effect of seasonality on precision and recall for Lisbon (blue) and Munich (green)

was performed to ensure comparability among the results. To compare the performance a precision recall plot was used (Saito and Rehmsmeier, 2015). Recall and precision were calculated for each of the four seasons (MAM, JJA, SON, DJF) and for the two half years (MAMJJA and SONDJF) using the estimated scaling factors. Results of the calculation are shown in Fig.

5. The dotted line is marking the line under which the classifier shows no skill. The line is defined as a proportion of drought events against overall amount of events (Saito and Rehmsmeier, 2015). For the Lisbon domain it becomes evident that the model performance is very different across seasons: higher precision of around 0.23 can be found during the winter half year. However for the spring season and summer halfyear the recall rises, while precision goes down. For the Munich classifier the results for the different seasons are closer together in terms of recall. It shows a worse performance for the winter months

(DJF), while fall, spring and summer show an overall better model performance. This is an indication that for the Munich domain better drought predictability is possible in spring, fall and summer.

An additional analysis is conducted to calculate the Shapely values separately for the four season and the two domains in order to understand the influence of the different variables on the prediction. The results of the analysis can be seen in Fig. 6 and 7. The results for the Lisbon domain show that NAO1 is the strongest predictor in winter and spring season, while

the contribution of pressure on drought predictability is higher in fall, followed by NAO1. On the contrary for the summer season NAO1 is not among the top 10 predictors, but other teleconnection indices like EAWR5, NAO7 and SCA7. Those teleconnection indices are originating from winter months where NAO showed to have the highest impact on the prediction. However, given the low performance of the model in the summer season, further investigation is needed. For the Munich domain NAO1 has one of the highest contributions for spring, summer and fall, while it cannot be found among the strongest





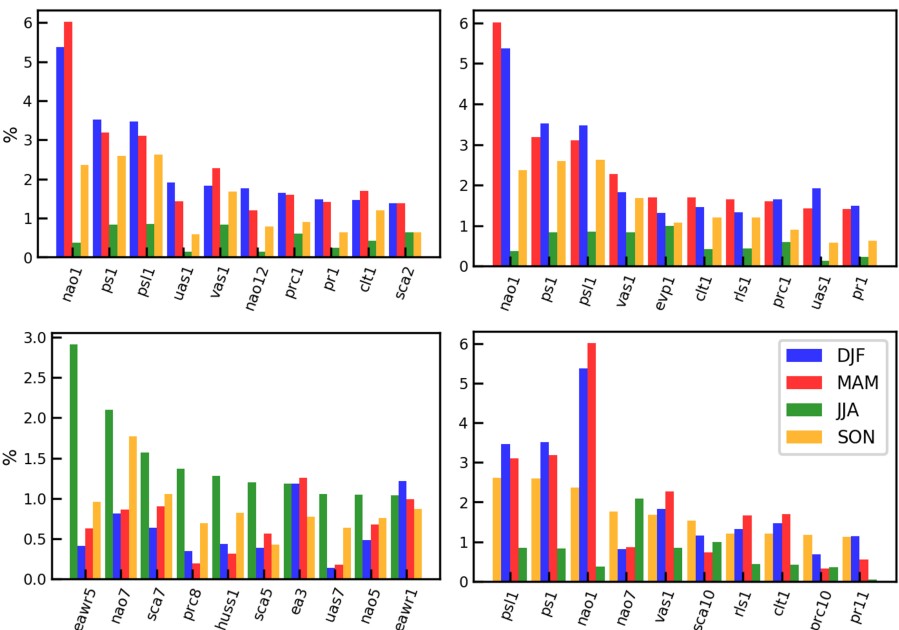

**Figure 6.** Shapely values for Lisbon calculated separately for the four seasons and sorted by the maximum contribution in DJF (top left), MAM (top right), JJA (bottom left) and SON (bottom right). evpsbl abbreviated as evp.

predictors for winter. EAWR5 is one of the strongest predictors for summer, spring and fall. The feature contributions for predictions in the winter season in Munich indicate that atmospheric variables 10 or 12 month before the event might be drought indicators.

## 4    Discussion and conclusion

Drought is a multiscale phenomenon and its formation and evolution is different to every climatology and season. In this
study, we i) explored the possibilities of using the data provided by CRCM5-LE to predict droughts using ANN and ii) applied explainable AI methods to gain a better understanding of the results. A drought event was defined as a SPI1 less than -1 at the given site. The first half of the study dealt with the systematic search for best performing models. For the Lisbon domain the model with L2-Regularization of 0.001, mean absolute error as loss function and the following architecture obtained best results: 5000*0.5*4000*0.5*1000*500*100. For the Munich domain the model with L2-Regularization of 0.001, mean
absolute error as loss function and the following architecture obtained best results: 4000*0.5*1000*0.5*500*100*5. Best performing models obtained accuracies of 57% for the Lisbon domain and 55% for the Munich domain.

     The precision of the prediction in both cases was rather moderate, as a high percentage of data is misclassified. For Lisbon, classifier precision remains at around 22 %. This means that one out of four predicted drought events is an actual drought. For the Munich case, this ratio is even lower and amounts to 18 %. However, the models provide an important basis for the



Natural Hazards
and Earth System
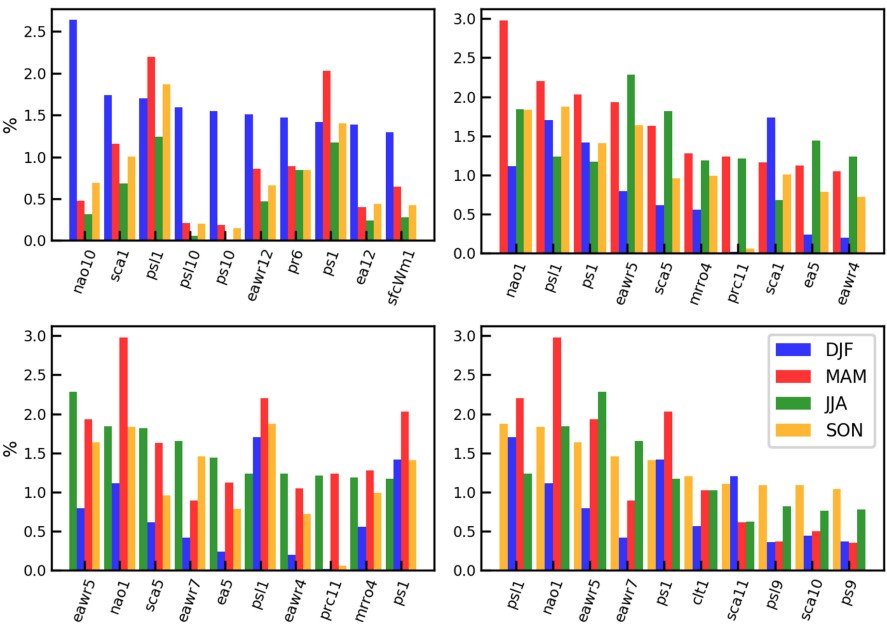

**Figure 7.** Shapely values for Munich calculated separately for the four seasons and sorted by the maximum contribution in DJF (top left), MAM (top right), JJA (bottom left) and SON (bottom right). sfcWindmax abbreviated as sfcWm.

development of future drought predicting models and offer a fruitful ground for the investigation of influence of single input variables during different seasons on drought formation.

     Compared to the study by Santos et al. (2014), which investigated drought predictability in Portugal, the weak prediction accuracies of our study are not surprising. SPI6 for April, May and June was predicted, however precipitation amounts for the months until March were also given as input. As SPI6 is calculated using the sum of 6 months precipitation, the model is

receiving over the half of the information it needs for the calculation of the value. As no similar studies exist for the Munich domain, no comparison can be performed.

     The second half of the study concentrated on the analysis of the obtained algorithms using explainable AI methods. Among the strongest predictors for the domains were NAO, *psl* and *ps* one month before the event. This underlines the importance of the atmospheric system on the drought formation. For the model trained for the Lisbon domain the variables of Northward

Near-Surface Wind (*vas*) and Evaporation (*evspsbl*) followed. For the Munich domain, EAWR and SCA five month before the event were found among the strongest predictors. In general the percentages of the contribution of the strongest predictors for the Munich domain were around six times lower than those for the Lisbon domain.

     This study indicates that seasonality is a crucial factor for drought predictions. Precision and recall of the prediction is getting lower in summer for the Lisbon domain and for winter for Munich domain. Moreover while for Munich domain the

spread of precision and recall across the seasons is rather low, huge differences were found for Lisbon domain: the trained model obtained higher recall and lower precision for spring and higher precision and lower recall for fall when comparing to



the baseline of all data. The results showed that for the Lisbon domain NAO1 is the strongest predictor in winter and spring season, while the contribution of pressure on drought predictability is higher in fall, followed by the contribution of NAO1. For Munich domain NAO1 was found to have one of the highest contributions for spring, summer and fall, while it could not be found among the ten strongest predictors for winter.

Further investigations are of interest for scientific research on both objectives. In terms of drought prediction, further research is possible within the same setting. The field of AI is evolving rapidly, showing new algorithms, methods and frameworks, such that there is a high potential for finding better suited algorithms (Hao, 2019). Given the high Shapely importance of NAO for drought prediction, other large scale variables, such as atmospheric blocking, can be added to the input variables. Moreover, the application to new domains is of interest to investigate the regionality of drought prediction possibilities. Explainable AI methods offer an important approach to improve the current limitations of machine learning models; their application is of high importance in the field of physical geography since it enables providing a physical interpretation to statistical results.

*Data availability.* Ensemble model data used in this study may be retrieved from the following sources: CanESM2-LE data are available via https://open.canada.ca/data/en/dataset/aa7b6823-fd1e-49ff-a6fb-68076a4a477c (Environment and Climate Change Canada, 2020). CRCM5-LE data can be retrieved at https://climex-data.srv.lrz.de/Public/ (Ouranos, 2020). The ERAInterim reanalysis data set was obtained at https://apps.ecmwf.int/datasets/data/interim-full-daily/levtype=sfc/ (European Centre for Medium-Range Weather Forecasts, 2020).

*Author contributions.* This study was conceptualized by EF under supervision of RL. Formal analysis, visualization of results and writing of the original draft was performed by EF. All authors contributed to the interpretation of the findings and revision of the paper.

*Competing interests.* The authors declare that they have no conflict of interest.

*Acknowledgements.* The CRCM5-LE was created within the ClimEx project, which was funded by the Bavarian State Ministry for the Environment and Consumer Protection. Computations of the CRCM5-LE were made on the SuperMUC supercomputer at Leibniz Supercomputing Centre of the Bavarian Academy of Sciences and Humanities. We acknowledge Environment and Climate Change Canada for providing the CanESM2-LE driving data.



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
