# Peer review of "Applying machine learning for drought prediction in a perfect model framework using data from a large ensemble of climate simulations"

_Natural Hazards and Earth System Sciences, 2021_

## Author Comment (AC1)

Authors reply to anonymous referee #1 are provided point by point in blue characters.

**Comment on nhess-2021-110**

Anonymous Referee #1

Referee comment on " Applying machine learning for drought prediction using data from a large ensemble of climate simulations" by Elizaveta Felsche and Ralf Ludwig, Nat. Hazards Earth Syst. Sci. Discuss., https://doi.org/10.5194/nhess-2021-110-RC1, 2021

**Title:** Applying machine learning for drought prediction using data from a large ensemble of climate simulations
**Author(s):** Elizaveta Felsche et al.
**MS No.:** nhess-2021-110
**MS type:** Research article: First review
**Special Issue:** Recent advances in drought and water scarcity monitoring, modelling, and forecasting (EGU2019, session HS4.1.1/NH1.31).

**RC1**: 'Comment on nhess-2021-110',

As presented by authors this study consists of two parts:

the first part focuses on a systematic **search** for the best performing setup of ANN models for Munich and Lisbon.

the second part focuses on the **analysis** the best performing models using explainable AI methods.

The set-up of this paper seems to have significant problems as both the search and the analysis have been performed using the same data-set (TEST – set) which actually a very small data set (years 2000-2005, only five years of monthly data for SP1 case).

The authors are using a large ensemble of 50 members for the study. This means that the period 2000 – 2005 is available 50 times, resulting in 250 model years for the test dataset. The same is true for the training dataset: There the period of 1957-1999 is to be multiplied by 50, resulting inf 43*50=2150 model years. The authors apologize for the misleading description and will work on clarifying this in the manuscript.

The authors should have used a hold-out set to investigate the actual performance in "unseen" data.

The authors consistently used "unseen" test data to evaluate the performance of the algorithms. During the training of the algorithms an additional validation set was selected from training data, to prevent overfitting and monitor the performance. A similar methodology was used e.g. in Morid et al. (2007).

In any case both the architecture selection, loss functions performance appear with really low F1 score values (below 0,3 in test set) – whereas the authors have stated earlier "we require that the accuracy on each class is at least 50%".

Authors thank the reviewer for the comment. The accuracy on both classes is at least 50%, however, due to the class imbalance the marginal F1 Score is low. An explanation to this behavior is given in P.

9, L. 180-182: "Due to the class imbalance within the dataset we require that the accuracy on each class is at least 50%. In that case given the distribution of the test dataset of 1803 non-drought events to 387 droughts for Lisbon and 1848 non-drought events to 352 drought events for Munich a marginal F1-score of 0.26 for Lisbon and 0.24 for Munich is given."

The paragraph that presents the model architecture is not clear. How many layers and neurons do we have in the selected model(s)?

The model architecture consists of overall seven layers; two of those are Dropout Layers, which are setting in a random way half of the neuron outputs to zero. Five layers are Dense neuron layers. The architecture of the layers is given in Table 5. For example the architecture for the model mentioned in the first line of Table 5 is the following:

1. Dense layer with 4000 neurons
2. Dropout Layer randomly setting 50% of weights to zero
3. Dense layer with 1000 neurons
4. Dropout Layer randomly setting 50% of weights to zero
5. Dense layer with 500 neurons
6. Dense layer with 100 neurons
7. Dense layer with 5 neurons

The authors apologize for the misleading description and will work on clarifying this in the manuscript.

This performance cannot and should not be considered as appropriate for a forecasting model. Therefore, both models (Lisbon and Munich) cannot be used for drought prediction.

This is something that the authors actually acknowledge as they state "The precision of the prediction in both cases was rather moderate, as a high percentage of data is misclassified".

Authors thank the reviewer for the comment and will add it to the limitations in the revised manuscript version.

The second half of the study presents the analysis of the performance obtained architectures.

In the 3.2.1 Shapely values section, we do not know which data set has been used – we assume that we are looking at the Test set. For Lisbon the cumulative contribution of the top 15 variables (out of the 27) is 20% which should explain the underperformance of the selected architecture. The case of Munich is even worse as the cumulative contribution of the top 15 variables is less than 5%.

Similar performance can be seen with seasonality analysis.

The analysis was performed on the test dataset. This will be added to the revised version of the manuscript.

A series of twelve months of each variable was taken as input to the machine learning model. For the calculation of Shapely values each month of each variable was considered individually, resulting in 27 atmospheric variables *12 + 6 teleconnection indices * 12 = 396 variables. This means that the cumulative contribution of the top 15 variables out of 396 variables (not 27) amounts to 20%/5% for

Lisbon/Munich. Authors thank the reviewer for the comment and will add the above explanation to the revised manuscript version.

The authors are aware of the limitations and would argue that although there is a comparably weak performance the obtained results can be of huge value for the development of a forecasting model.

Last, in conclusion (line 290) the author state "Best performing models obtained accuracies of 57% for the Lisbon domain and 55% for the Munich domain". This is not true as this performance has been seen in train set , not the test set. Even if it was in the test set it would have been insufficient as the model has already seen the information in the data set and therefore should not be considered for forecasting performance evaluation

The stated performance was seen on the test dataset, which was not used for model training, therefore the authors argue that the result can be used for the forecasting performance evaluation. In P. 10 L. 210-212 the authors state that only the results on the training set are shown: ". Training results are displayed in this particular case [L2 regularization] since the regularization is introduced to prevent overfitting. Generally the performance on the test dataset is more important and will be inspected in following experiments." The authors apologize for the misleading description and will work on clarifying this in the manuscript.

---

## Author Comment (AC2)

**Comment on nhess-2021-110**

Anonymous Referee #2

Referee comment on " Applying machine learning for drought prediction using data from a large ensemble of climate simulations" by Elizaveta Felsche and Ralf Ludwig, Nat. Hazards Earth Syst. Sci. Discuss., https://doi.org/10.5194/nhess-2021-110-RC2, 2021

**Title:** Applying machine learning for drought prediction using data from a large ensemble of climate simulations
**Author(s):** Elizaveta Felsche et al.
**MS No.:** nhess-2021-110
**MS type:** Research article: First review
**Special Issue:** Recent advances in drought and water scarcity monitoring, modelling, and forecasting (EGU2019, session HS4.1.1/NH1.31).

RC2: 'Comment on nhess-2021-110',

This study present a methodology for drought prediction at seasonal scale using machine learning algorithms. The study is treating a highly relevant subject with the usage of a novel methodology based on machine learning for droughts predictions. However, the issue with machine learning and climate is the length of the climate records, which does not allows to build AI models. Therefore, this study proposes a methodology fully based on a down scaled ESM. The study is interesting however, I have major comments listed bellow, in particular, at that stage it is very hard to evaluate properly the manuscript since the data/method is not clear enough:

1) the method (if I understood it correctly) is fully based on model data, therefor it is not really a study about prediction but according to me it is only potential predictability, since this study does not demonstrate any skill in predicting observed past climate in the two regions of interest, but only the ability to forecast the model climate. The paper should be much clearer about this, for example the title and the abstract should use the term "perfect model framework" and/or potential predictability.

Authors thank the reviewer for his/her useful suggestion. The title of the revised version of the manuscript will include it:
"Applying machine learning for drought prediction in a perfect model framework using data from a large ensemble of climate simulations"

2) The method description is very unclear about the prediction aspects. What are the target month analyzed? From which start date? For example, it is really confusing to me to predict SPI1 with one month lead time for different seasons. What do you mean here? Do you mix all together the start date of March (to predict the SPI1 of April), April (to predict the SPI1 of May) and May (to predict the SPI1 of June)? Or do you predict SPI1 integrated over MAM, but in this case, to my understanding it is not SPI1 but SPI3. In any case the methodology should be much clearer about this point, at this stage I cannot evaluate properly the manuscript without this clarification.

The authors apologize for the misleading description and will work on clarifying this in the manuscript. The study predicts SPI1 with a lead time of one month. To predict e.g. SPI1 in April of 2000, the data for twelve months before the event is used as input, this is SPI1 and other variables for the period April 1999 – March 2000. For the calculation of SPI1 in April only precipitation for the month of April is used.

3) "The data from the years 1957 - 1999 was used as training data, the years 2000-2005 were used for the testing purpose." Do you mean that the score calculation is performed only for 6 years from 2000 to 2005? This is a far too short period for any skill assessment. Usually, in seasonal prediction the skill is assessed over the whole hindcast period (1957-2005), using cross validation to construct the prediction.

We are using a large ensemble of 50 members for the study. This means that the period 2000 – 2005 is available 50 times, resulting in 250 model years for the test dataset. The same is true for the training dataset: There the period of 1957-1999 is to be multiplied by 50, resulting inf 43*50=2150 model years. The authors apologize for the misleading description and will work on clarifying this in the manuscript. The authors are aware that cross validation would add value to the study, however given the huge amount of variables and the fact that the overall analysis includes 2500 model years, it would require huge computational resources, that are not available.

4) The discussion does not mention at all the main limitation of this study according to me: at that stage the authors have demonstrated some ability to predict a model using AI, but we don't know how to use such method for real prediction. Would it be possible to apply your model on observation and then verify its skill? If yes, it should be included in the study and if not this should be clearly mentioned.

Authors thank the reviewer for his/her useful suggestion. The authors would argue that the immediate application of the framework on observation is not possible, due to the fact that observational data usually lacks a multitude of variables which were used as input in this study e.g. Heat Fluxes, radiation, etc. The objective of this study was not to develop a framework that can be applied on observation, but to use the large amount of events provided by the large ensemble for prediction. The results obtained by shapely value calculation are of high importance for the choice of variables for a development of a model which could be applied to observational data.

Typos:
This study uses the monthly sea level pressure (pr)

The stong influence of ps/psl and NAO shows the influence of the atmospheric pressure

Typos will all be fixed in the preparation of the revised manuscript version.

---

## Author Response (AR2)

We warmly thank the editor and the referees for carefully reading the manuscript and their valuable comments. The author's answers are provided point by point in blue characters.

**Comment by the editor**

I have reviewed the latest revised version of the paper, the reviewers' comments. The authors have addressed most of the reviewers' comments during the first review round. However, I would agree with the comments made by reviewer#3 during the second review round. The revisions requested by the reviewer are quite moderate and valid. Reviewer #2 has accepted the paper for publication. I suggest that the authors address the revisions required by reviewer#3 and send the revised paper for a final review by the editor and the final decision.

**Report #1 by Anonymous Referee #3**

The paper deals with an important issue in the field of changing climate and the use of machine learning methods to identify drought conditions. In my opinion, the paper has been prepared in a good manner and presents adequate technical and experimental details. The paper is a novel study and is generally well-structured as it explains the methodology, the mathematical framework and the assumptions used, and the justification of the results and the conclusions. However, the application research part needs improvements and corrections to verify the novelties of the method employed in the study area. Furthermore, there are few critical points that should be addressed in the revised manuscript. Addressing these comments will improve the quality of the paper and help the general reader of the paper. The paper could be accepted for publication considering the following revisions.

1. Why the authors use the parametric Pearson correlation ($\rho$) and not a non-parametric test (like Kendall tau and/or Spearman rho)? In case of nonlinear correlation between climatic signals and local drought values, statistics such as mutual information (MI) could be more informative than the conventional correlation coefficient. Therefore, I suggest to present mutual information as another statistic in Table 1. The authors state (lines 116-117) "The correlation coefficients reveal that out of the full list of 42 variables 14 are sorted out as being redundant". Hence, nonlinear and/or non-parametric statistic values should be added to verify this conclusion.

The authors thank the reviewer for the helpful suggestion. An explanation for the necessity of variable subset selection is added in the L. 107-108. The authors prefer not to include mutual information for the following reason:
As this calculation step aims to omit redundant variables, we are interested in the upper bound of the value. By definition, Mutual Information (MI) has no upper bound (Strehl et al., 2002). Therefore, MI is not comparable between the different variables. A normalized version of MI by Strehl et al. (2002) then approximates MI akin to the Pearson correlation coefficient.

2. Justification of the selected timescale of SPI and the selected forecasted lead-time. Why the authors use the SPI-1month?
Due to the overall complexity of the input dataset with 28 atmospheric and soil variables and 2500 model years that are used for the analysis, our goal was to find a robust setup in terms of lead-time to explore the influence of the prediction variables on the prediction. As shown by previous studies, shorter prediction lead times are usually more robust than longer periods (Belayneh et al., 2012). To calculate SPI3/SPI6, precipitation values for the preceding three/six month months are used. As noted in Yoon et al. (2012), when performing a prediction of a lead-time less than the accumulation period of the SPI value, the skill of the forecast can largely be explained due to this relationship.

Therefore, the authors argue that the accumulation period should not be chosen any lesser than the lead time to evaluate the effects besides the explained relationship. Therefore the authors chose to use a lead-time and accumulation period of one month. One month lead-time was also used in previous studies by Yoon et al. (2012) and Deo et al. (2017). The authors have added the explanation to the revised version of the manuscript.

How reliable is the SPI-1 for drought prediction at the study sites?

The two study sites have hugely different meteorological conditions, especially in terms of precipitation averages throughout the year. While Lisbon has a Mediterranean climate, Munich has a continental one. As noted by Zargar et al. (2011), SPI is essentially a measure to compare the precipitation departure from normal, and therefore it is a measure that applies to highly different climates and makes them comparable. As the complexity and related uncertainties of the prediction rise with extended lead times, the authors chose a lead time of one month. Therefore, the accumulation period of SPI had to be also chosen one month, as explained in the previous answer. Unfortunately, no comparable studies exist for the two domains to be able to evaluate the prediction performance.

Why the previous 12 months are used as input for predicting the SPI of the next month (please provide scientific evidence for this assumption and why not testing up to 36 months before)?

The twelve months before the event are chosen following the study by Morid et al. (2007), which found that the best performing drought prediction model was the one including the value up to twelve months before the predicted one. The explanation has been added to the revised version of the manuscript in L. 150-152. The authors are aware that experimenting with enlarging the input period might improve the algorithm's performance. However, given the huge amount of variables and the fact that the overall analysis includes 2500 model years, it would require substantial computational resources.

3. The results clearly show (see Accuracy and F1-score) that the final selected models cannot be used for forecasting purposes for the selected drought index. The Threat Score (TS) or the Critical Success Index (CSI) could be used to verify if the proposed method should be used (https://www.cawcr.gov.au/projects/verification/Hewson/DeterministicLimit.html). Furthermore, I have the impression that the selected timescale of SPI (1-month) leads to unstable results. If the authors use for example the SPI-6 month how different would be the derived results?

As described in comment 2), the authors would expect the performance to improve for the prediction of SPI6 with a lead time of one month due to the fact that five out of six values needed for a numerical calculation of SPI6 would be given as input to the algorithm. Therefore, the study's approach was to display the dependencies beyond the ones given by the definition of the drought index.

The authors thank the reviewer for the useful suggestion to introduce additional verification by calculating the Threat Score (TS)/Critical Success Index (CSI). However, the authors would argue in accordance with Jolliffe et al. (2002) that the score is less suitable for our problem, as it is highly dependent on the frequency of the event and therefore biased. Instead, the Heidke Skill Score (HSS) is proposed as an unbiased version given the high class imbalance. For the best-performing models, HSS equals 0.06 for Lisbon and 0.04 for Munich. These results confirm that the obtained prediction is better than the random forecast and therefore show a weak prediction skill. The HSS is added to the revised version of the manuscript.

Minor comments

1. A flow chart of the proposed method may also be added. The authors are requested to ensure that international readers/scientists will be able to apply this methodology on their data sets by following the flow chart.

Authors thank the reviewer for the useful suggestion. A flowchart is added to the revised version of the manuscript.

**References**

Belayneh, A., Adamowski, J., Khalil, B., and Quilty, J.: Coupling machine learning methods with wavelet transforms and the bootstrap and boosting ensemble approaches for drought prediction, Atmospheric Research, 172, https://doi.org/10.1016/j.atmosres.2015.12.017, 2016.

Deo, R. C., Kisi, O., & Singh, V. P. (2017). Drought forecasting in eastern Australia using multivariate adaptive regression spline, least square support vector machine and M5Tree model. Atmospheric Research, 184, 149-175.

Jolliffe, I. T., & Stephenson, D. B. (2012). Forecast verification: A practitioner's guide in atmospheric science. Chichester, West Sussex: Wiley-Blackwell.

Morid, S., Smakhtin, V., and Bagherzadeh, K.: Drought forecasting using artificial neural networks and time series of drought indices, INT J CLIMATOL, 27, 2103–2111, https://doi.org/10.1002/joc.1498, 2007.

Strehl, A., & Ghosh, J. (2002). Cluster ensembles---a knowledge reuse framework for combining multiple partitions. Journal of machine learning research, 3(Dec), 583-617.

Yoon, J. H., Mo, K., & Wood, E. F. (2012). Dynamic-model-based seasonal prediction of meteorological drought over the contiguous United States. Journal of Hydrometeorology, 13(2), 463-482.

Zargar, A., Sadiq, R., Naser, B., & Khan, F. I. (2011). A review of drought indices. Environmental Reviews, 19(NA), 333-349.